# An approach in medical diagnosis based on Z-numbers soft set

**Haiyan Zhao[1], Qian Xiao[1,2]\*, Zheng Liu[1], Yanhong Wang[1]**

**1** School of Management, Shanghai University of Engineering Science, Shanghai, P.R. China, **2** School of Economic and Management, Shanghai University of Finance and Economics, Shanghai, P.R. China

\* 817589@qq.com

## Abstract

### Background

In the process of medical diagnosis, a large amount of uncertain and inconsistent information is inevitably involved. There have been many fruitful results were investigated for medical diagnosis by utilizing different traditional uncertainty mathematical tools. It is found that there is limited study on measuring reliability of the information involved are rare, moreover, the existed methods cannot give the measuring reliability of every judgment to all symptoms in details.

### Objectives

It is quite essential to recognize the impact on the reliability of the fuzzy information provided under inadequate experience, lack of knowledge and so on. In this paper, the notion of the Z-numbers soft set is proposed to handle the reliability of every judgment to all symptoms in details. The study in this paper is an interdisciplinary approach towards rapid and efficient medical diagnosis.

### Methods

An approach based on Z-numbers soft set (ZnSS)to medical diagnosis has been developed and is used to estimate whether two patterns or images are identical or approximately. The notion of Z-numbers soft set is proposed by combing the theory of soft set and Z-numbers theory. The basic properties of subset, equal, intersection, union and complement operations on the Z-numbers soft sets are defined and the similarity measure of two Z-numbers soft sets are also discussed in this paper.

### Results

An illustrative example similar to existing studies is showed to verify the effectiveness and feasibility, which can highlight the proposed method and demonstrate the solution characteristics.

**Data Availability Statement:** All relevant data are within the paper and its Supporting information files.

**Funding:** LZ is funded by the National Social Science Fund of China (Grant No. 18CGL015). YW is funded by the Humanity and Social Science foundation of Ministry of Education of China (Grant

No. 20YJC630150). The funders had some role in study design, data collection and analysis, decision to publish, or preparation of the manuscript.

**Competing interests:** The authors have declared that no competing interests exist.

## Conclusion

Diagnosing diseases by uncertainty symptoms is not a direct and simple task at all. The approach based on ZnSS presented in this paper can not only measure reliability of the information involved, but also give the measuring reliability of every judgment to all symptoms in details.

## 1. Introduction

Medical diagnosis has been considered as one of the most important and crucial processes which determines diseases of patients by some given symptoms from the certain observers [1]. In the process of medical diagnosis, a large amount of uncertain and inconsistent information is inevitably involved [2]. General uncertainty (vagueness or imprecision) arises due to patients' vague linguistic expression of their problems to medical experts. Furthermore, some sort of uncertainty may occur due to deficiencies in lab experiments and human error [3]. Due to the information available for experts about the patient's symptoms in general is inherently uncertain. Research on the theories and methodologies of decision making for real world problems under uncertainty are important and necessary [4–8]. Diagnosing diseases by uncertainty symptoms is not a direct and simple task at all and handing uncertainty in medical diagnoses is an open issue.

One of the challenges in the process of this kind medical diagnosis is about how to handle the uncertainty effectively to achieve more accurate decision-making. As a matter of fact, these traditional uncertainty mathematical theories have developed greatly and achievements have been widely applied in medical diagnosis fields [9–18]. In the past decade, many researches were investigated for medical diagnosis by utilizing different technologies, like intuitionistic fuzzy sets [9], interval-valued intuitionistic fuzzy sets [11, 12], rough sets theory [12], evidential reasoning [13], quantum decision [14], machine learning [15, 16], and fuzzy sets theory [17, 18]. However, in the process of medical diagnosis, there may be many critical diseases, where experts do not have sufficient knowledge to handle those problems. For these cases, experts may provide their opinion only about certain aspects of the disease based on the symptoms they focused and remain silent for those unknown symptoms. For this kind of situation, the traditional uncertainty mathematical theories mentioned above have their intrinsic difficulties [19], which are pointed by Molodtsov. Molodtsov initiated the theory of soft sets as a nonparametric way for handling vagueness and uncertainties, which has been proven useful in many fields. Theory of soft sets has inherent advantage for dealing with uncertainties which traditional mathematical tools cannot handle.

Therefore, one purpose of this paper is to apply the advantage of soft set to deal with uncertainty to solve this kind of special situation in medical diagnosis. For one aspect, it is free from the inadequacy of the parameterization tools of traditional mathematical tools [20]. For another aspect, theory of soft sets can be used by combining other traditional uncertainty mathematical tools. So far, many mathematical theories combined soft sets and other uncertainty tools, such as fuzzy soft set [21], vague soft set [22], soft rough sets [23], intuitionistic fuzzy soft sets [24], fuzzy hyper soft sets [25], interval-valued soft set [26] and interval-valued intuitionistic fuzzy soft sets [27], multi-fuzzy N-soft [28–31], have been employed as effective and useful tools to deal with uncertainties. Further researches on the diverse soft sets mentioned above have also been conducted into medical decision-making problems [32–40]. Theory of Fuzzy soft sets are applied in the field of medical decision making at the earliest [32]. As

practical applications, the intuitionistic fuzzy soft sets have been successfully applied into medical system [37], and group decision making method using intuitionistic fuzzy soft sets is of great significance in aiding medical diagnosis [40]. These rich research methods provide reference and support for this paper.

Another challenge in the process of this kind medical diagnosis is about how to measure reliability of the information effectively to achieve more accurate decision-making. For instance, to describe the uncertainties of the kind of "fever will surely be very high", membership degree of fuzzy sets, even non-membership degree of intuitionistic fuzzy sets can express the fuzzy uncertain concept "high fever" better, but the fuzzy uncertain concept "surely" cannot be model. Obliviously, there are many such kind inherently uncertain descriptions due to the information available for observers or experts about the patient's symptoms in medical diagnosis. There have been many fruitful results combining soft set and traditional uncertainty mathematical tools, we found that studies on measuring reliability of the information involved based on soft sets are rare. To some extent, the concept of level-fuzzy soft set [21], intuitionistic level-fuzzy soft set [21, 40, 41], generalized fuzzy soft set [32] and generalized intuitionistic fuzzy soft sets [42] can deal with certain reliability. The application of level soft sets can partly express the reliability of judgment on information, such as in reference [43]. Sujit Dasa and Samarjit Kar presented a confident weigh way to each expert in group decision making problem in medical diagnosis [32]. However, these methods only can give whole reliability to all attributes, without giving the measuring reliability of every judgment or information in details. Z-numbers were proposed by Zadeh as a new way to deal with uncertainty and reliability of information [44]. Z-numbers are composed of two parts: one part denotes a restriction on values that can be assumed, and another part is the reliability of the information. Z-numbers may be used to model uncertainty description like "fever with pneumonia will surely be very high", "headache with cold will surely be very heavy", "cough with influenza will high probability be severe", etc.

Obviously, Z-numbers can describe levels of human judgment in details. Z-numbers have a significant potential in the describing of the uncertainty of the human knowledge because it consists of restraint and reliability of the measured value in details. Given the advantages of Z-numbers in describing uncertain information, it has been researched extensively and applied widely in various fields in recent years. Theoretical research on the Z-numbers continues comparatively active, such as the arithmetic of different Z-numbers [45–48], the ranking of Z-numbers [49–51], the utility of Z-numbers [52], the Z-valuations and function of Z-numbers [53, 54], etc. And that applied research on the Z-numbers continues to be scare but growing fast, such as Measuring Uncertainty [55], decision-making Analysis [56–58], group decision-making problem [59–62], optimization [63], medical diagnosis [64], Control System Design [65], etc.

Noting that the soft set provides a technological tool to extend the concept of Zadeh's fuzzy set, while Z-numbers is a conceptual extension of a fuzzy set. As mentioned above, both of soft sets and Z-number have been widely applied to realistic areas, but the combination of Z-number and fuzzy soft set shows the theoretical applications of the two models. Motivated by the ideas of fuzzy soft sets, a direct extension would be combining soft set and Z-numbers. Therefore, in this paper, we try to conduct Z-numbers soft set to deal with nonparametric uncertainty and reliability of information by combining the advantage of the Molodtsov soft set theory and Zadeh's Z-numbers concept, which is the core purpose of this paper. Hence, we take advantage of Z-numbers and soft set by combing them. Moreover, the basic operations and properties of Z-numbers soft sets also are defined, and approach to medical diagnosis has been developed based on similarity measure on two Z-numbers soft sets. The approach based on ZnSS presented in this paper can not only measure reliability of the information involved,

but also give the measuring reliability of every judgment to all symptoms in details. It is important to note that professional medical knowledge is not used in this paper.

Consequently, the contribution and originality of this study are summarized as follows:

1. The notion of the Z-numbers soft set is proposed by combing the theory of soft set and Z-numbers in this paper. Z-numbers soft set has the desirable merit which can take advantage both of soft set and Z-numbers, which is capable of solving the reliability of every judgment to all symptoms in details.

2. The subset, equal, intersection, union and complement operations are also defined on the Z-numbers soft sets. The basic properties of the Z-numbers soft set are also presented.

3. It is required to compare two Z-numbers soft sets. Hence, in this Paper a measure of similarity between two Z-numbers soft sets has been given and discussed.

4. An novel approach to medical diagnosis has been developed based on similarity measure of Z-numbers soft sets. The approach based on Z-numbers soft set can measure reliability of every judgment to all symptoms effectively to achieve more accurate decision-making result in medical diagnosis.

To realize the contributions mentioned above, the remainder of this paper is organized as follows. Section 2 reviews some basic definitions and concepts are used in the following sections. In section 3, we investigate the theory of Z-numbers soft set. Section 4 presents an approach to decision-making based the similarity measure of Z-numbers soft set. In section 5, an illustrative example in medical diagnosis to show that the proposals presented in this paper are not only more reasonable but more efficient in practical application. At last, we conclude our analysis and set out further research directions in Section 6.

## 2. Preliminaries

In this section we give a briefly review of some basic definitions and properties used in the following sections. If we use membership function (characteristic function or discrimination function), we can represent whether an element $x$ is involved in a set $A$ or not.

Let $U$ be a non-empty finite universe. For any $A \subseteq U$, $A(x) = \left\{ \begin{array}{l} 1, x \in A \\ 0, x \notin A \end{array} \right\}$ is a characteristic function that is a mapping from $U$ to $\{0, 1\}$. Then any characteristic function on universe $U$ determines a classical crisp set of $U$, that is $A = \{x \in U mid A(x) = 1\}$.

Consequently, fuzzy set is 'vague boundary set' comparing with crisp set. The fuzzy set concept presented by Zadeh provides a framework to expressing vague concepts by allowing partial memberships.

### Fuzzy set

Definition: Fuzzy set [9]. Let $U$ be a non-empty finite universe. A fuzzy set $A = \{(x, \mu_A(x))\}$ on the universe $U$ is defined by the mapping $A = \{(x, \mu_A(x))\} \mu_A(x) : U \to [0, 1]$, where $[0, 1]$ means real numbers between 0 and 1 (including 0, 1); $\mu_A(x)$ denotes the membership degree. That is to say, each element is mapped to $[0, 1]$ by membership function.

There are lots of definitions for fuzzy operations. According to the max-min system given by Zadeh [9], the fuzzy intersection, union, and complement are defined as follows: $\mu_{A \cap B}(x) = \min\{\mu_A(x), \mu_B(x)\}; \mu_{A \cup B}(x) = \max\{\mu_A(x), \mu_B(x)\}; \mu_{A^c}(x) = 1 - \mu_A(x)$.

Fuzzy set can be used to describe perception and subjectivity of human as they represent uncertain or imprecise information. However, the reliability of information from different evaluator in the process of decision making such as medical diagnosis is also very important.

## Z-numbers

Z-numbers were proposed by Zadeh as a new way to deal with uncertainty and reliability of information [44].

Definition: Z-numbers2 [45]. A Z-numbers is denoted as $Z = (\tilde{A}, \tilde{B})$, which is an ordered pair of fuzzy numbers. $\tilde{A}$ is a fuzzy restriction on the values that a real-valued uncertain variable $X$ can take, and $\tilde{B}$ is a measure of reliability of $\tilde{A}$.

Hence, A Z-number is associated with a real-valued uncertain variable $x$. $\tilde{A}$ and $\tilde{B}$ are described in real number or natural language, such as (about 37.7 degree, quite sure). The ordered triple $(x, \tilde{A}, \tilde{B})$ is referred to a discrete Z-valuation [47], which can be understood that $x$ is $(\tilde{A}, \tilde{B})$, where $x$ is a variable, $\tilde{A}$ is a fuzzy set used to describe the restriction, and $\tilde{B}$ is a fuzzy number used by natural language to describe reliability of $\tilde{A}$. A discrete Z-valuation can be model sentences like "fever with pneumonia will surely be very high", where $x$ is a variable "fever", $\tilde{A}$ is a fuzzy set "very high", and $\tilde{B}$ is a fuzzy constriction used by natural language "surely" to describe reliability of $\tilde{A}$.

## Fuzzy soft set

In 1999, Molodtsov initiated soft set theory [19] as a new mathematical tool for coping with uncertainty that seems to be free from the inherent flaws. In this section $U$ refers to the initial non-empty universe of objects, cases, selections and so on. Let $E$ is a set of parameters and $A \subseteq E$, which are often attributes or characteristics of the objects. Let $P(U)$ denote the power set of initial universal set $U$, and a pair $(F, A)$ is called a soft set over $U$ iff F is a mapping given by $F$: $A \rightarrow P(U)$. Maji et al. [21] initiated fuzzy soft set, a more generalized notion combining both fuzzy sets and soft set.

Definition: Fuzzy soft set [21]. Let $E$ is a set of parameters and $A \subseteq E$. Let $(\tilde{P}, U)$ denote the power fuzzy set of initial universal set $U$, and a pair $(\tilde{F}, A)$ is called a fuzzy soft set over $U$ iff $\tilde{F}$ is a mapping given by $\tilde{F} : A \rightarrow \tilde{P}(U)$.

As an illustration, consider the following example [32].

Suppose a soft set $(F, A)$ describes diagnosis of diseases with respect to the given symptoms (parameters), which the duty doctor are going to get from the patient.

Let $U$ be the set of diagnosis of diseases under consideration($d_1$ = Viral fever, $d_2$ = Malaria, $d_2$ = Typhoid, $d_2$ = Gastric ulcer, $d_1$ = Pneumonia) given by $U = \{d_1, d_2, d_3, d_4, d_5\}$.

Also let $A \subseteq E$ be the set of symptoms ($e_1$ = Temperature, $e_2$ = Headache, $e_3$ = Stomach pain, $e_4$ = Cough, $e_5$ = Chest pain), given by $A = \{e_1, e_2, e_3, e_4, e_5\}$.

Let $(\tilde{F}, A)$ is a mapping given by $\tilde{F}(e_1) = \{\frac{d_1}{0.2}, \frac{d_2}{0.4}, \frac{d_3}{0.9}, \frac{d_4}{0.7}, \frac{d_5}{0}\}, \tilde{F}(e_2) = \{\frac{d_1}{0}, \frac{d_2}{0.8}, \frac{d_3}{0.1}, \frac{d_4}{0.7}, \frac{d_5}{0}\}, \tilde{F}(e_3) = \{\frac{d_1}{0.6}, \frac{d_2}{0.2}, \frac{d_3}{0.8}, \frac{d_4}{0}, \frac{d_5}{0}\}, \tilde{F}(e_4) = \{\Phi\}, \tilde{F}(e_5) = \{\frac{d_1}{0.6}, \frac{d_2}{0.7}, \frac{d_3}{0.5}, \frac{d_4}{0.8}, \frac{d_5}{0}\}.$

All the available information on these diseases can be characterized by a fuzzy soft set $(\tilde{F}, A)$. In reality, in many applications the related membership function can be confirmed but the reliability is extremely individual (depend on evaluator's judgment of attributes on alternatives) and thus cannot be lightly confirmed. P.K. Majumdar and S.K. Samanta [27] gave a modified definition of fuzzy soft set named generalized fuzzy soft set which can indicate not only the degree of belongingness but also the degree of possibility of such belongingness. However, it's more reasonable to give a degree of reliability on the evaluator's judgment of attribute

on alternatives but not whole reliability to all attributes. To improve classical fuzzy soft sets concepts, in this paper we propose the concept of Z-numbers soft set by combining both Z-numbers set and soft set.

## 3. On Z-numbers soft set

Definition: Z-numbers soft set. Let $U$ refers to the initial non-empty universe of objects, cases, selections and so on, and $E$ is a set of parameters and $A \subseteq E$. Let $ZnF(U)$ denote the power Z-numbers set of initial universal set $U$, and a pair $(\tilde{\tilde{F}}, A)$ is called a Z-numbers soft set over $U$ iff $\tilde{\tilde{F}}$ is a mapping given by $\tilde{\tilde{F}} : A \rightarrow \text{ZnF}(U)$, which can be detailed as Eq (1):

$$(\tilde{\tilde{F}}, A) = \{(x_i, (x_i \text{ satisfies parameter } e_j, A(x_i)_{e_j}, B(x_i)_{e_j})) : \forall x \in U, \forall e \in E\} \qquad (1)$$

where $A(x_i)_{e_j}$ is fuzzy restriction on the value of degree that alternative $x_i$ have parameter $e_j$ and $B(x_i)_{e_j}$ means the reliability of $A(x_i)_{e_j}$.

Here a Z-numbers soft set $(\tilde{\tilde{F}}, A)$ also can be noted $\tilde{\tilde{F}}_A$. Tao et al. [62] presented the concept of linguistic Z-numbers fuzzy soft sets and its application on a multi-attribute group decision making problem without any language transformation. In this paper Z-numbers soft set is shorted as ZnSS.

Actually, there are maybe several different kinds of Z-numbers soft set, such as continuous Z-numbers soft set, linguistic Z-numbers soft set, discrete Z-numbers soft set and so on. Because in a Z-numbers $(x, \tilde{A}, \tilde{B})$, the fuzzy number $\tilde{A}$ and $\tilde{B}$ may be the different fuzzy form such as continuous values, linguistic language value or some other discrete value. We can call them continuous Z-numbers, linguistic Z-numbers, discrete Z-numbers and so on. So, a Z-numbers soft set over $U$ can also be defined as Eq (2):

$$(\tilde{\tilde{F}}, A) = \{(x_i, (x_i \text{ satisfies parameter } e_j, \mu(x_i)_{e_j}, \gamma(x_i)_{e_j})) : \forall x \in U, \forall e \in E\} \qquad (2)$$

where $\mu(x_i)_{e_j}$ is fuzzy restriction on the value of degree that alternative $x_i$ have parameter $e_j$ and $\gamma(x_i)_{e_j}$ means the reliability of $\mu(x_i)_{e_j}$.

Undoubtedly, it can be converted to the value between [0, 1] no matter what kind of fuzzy form to use. Henceforth in this paper we will describe a Z-numbers set as $Z = (x, \mu_A(x), r_B(x))$ by using converted [0, 1] value of degree that alternative $x_i$ have parameter $e_j$ and the degree of reliability, that is to say $\mu(x_i)_{e_j} \in [0, 1], \gamma(x_i)_{e_j} \in [0, 1]$.

Obviously, every fuzzy soft set may be considered as a Z-numbers soft set when the reliability $\gamma(x_i)_{e_j}$ is constant one. A Z-numbers set could also be naturally viewed as a Z-numbers soft set whose parameter set is a singleton.

Here also consider an example just like example in section Preliminaries. Supposed all the available information can be characterized by a Z-numbers soft set $(\tilde{\tilde{F}}, A)$. Let $(\tilde{\tilde{F}}, A)$ is a

mapping given by $(\tilde{\tilde{F}}, A) = \{\tilde{\tilde{F}}(e_1), \tilde{\tilde{F}}(e_2), \tilde{\tilde{F}}(e_3), \tilde{\tilde{F}}(e_4), \tilde{\tilde{F}}(e_5)\}$, where,

$$\tilde{\tilde{F}}(e_1) = \{(d_1, 0.2, 0.5), (d_2, 0.4, 0.3), (d_3, 0.9, 0.6), (d_4, 0.7, 0.8), (d_5, 0, 0.8)\},$$

$$\tilde{\tilde{F}}(e_2) = \{(d_1, 0, 0.8), (d_2, 0.8, 1), (d_3, 0.1, 1), (d_4, 0.7, 0.1), (d_5, 0, 1)\},$$

$$\tilde{\tilde{F}}(e_3) = \{(d_1, 0.6, 0.5), (d_2, 0.2, 0.3), (d_3, 0.8, 0.5), (d_4, 0, 1), (d_5, 0, 0.9)\},$$

$$\tilde{\tilde{F}}(e_4) = \{\Phi\},$$

$$\tilde{\tilde{F}}(e_5) = \{(d_1, 0.6, 0.3), (d_2, 0.7, 0.6), (d_3, 0.5, 0.8), (d_4, 0.8, 0.5), (d_5, 0, 0.9)\}.$$

A Z-numbers soft set $(\tilde{\tilde{F}}, A)$ can also be represented in the form of a two-dimensional table, shown in Table 1.

Noting that if all the $\gamma(x_i)_{e_j} = 1$, the Z-numbers soft set $(\tilde{\tilde{F}}, A)$ is a fuzzy soft set $(\tilde{F}, A)$. If the $\gamma(x_i)$ for each $e_j$ is a constant value, the Z-numbers soft set $(\tilde{\tilde{F}}, A)$ is a generalized fuzzy soft set, which is presented by P.K. Majumdar and S.K. Samanta [27]. So, we can say fuzzy soft set and generalized fuzzy soft set both are different kind of Z-numbers soft set.

Definition: Z-numbers soft matrix. Let $(\tilde{\tilde{F}}, A)$ be a Z-numbers soft set over the initial universe $U$. Let $E$ be a set of parameters and $A \subseteq E$. Then a subset of $U \times E$ is uniquely defined by $\{(\mu(x_i)_{e_j}, \gamma(x_i)_{e_j}) : e \in A, x \in U\}$ which is called a relation $\hat{R}_A$ form of $(\tilde{\tilde{F}}, A)$. Now the relation $\hat{R}_A$ is characterized by the membership value $\mu(\chi_i)_{e_j}$ and the reliability degree value $\gamma(\chi_i)_{e_j}$. Then $\hat{R}_A$ is represented by a matrix as Eq (3):

$$\hat{R}_A = \begin{pmatrix} a_{11} & a_{12} & \cdots & a_{1n} \\ a_{21} & a_{22} & \cdots & a_{2n} \\ \vdots & \vdots & \ddots & \vdots \\ a_{m1} & a_{m2} & \cdots & a_{mn} \end{pmatrix} \tag{3}$$

where $a_{ij} = (\mu(x_i)_{e_j}, \gamma(x_i)_{e_j})$. The above matrix is called a Z-numbers soft matrix of order $m \times n$ corresponding to the Z-numbers soft set $(\tilde{\tilde{F}}, A)$ or $\tilde{\tilde{F}}_A$.

**Table 1. A ZnSS model for influenza $(\tilde{\tilde{F}}_A)$.**

| $U$ | $e_1$ | $e_2$ | $e_3$ | $e_5$ |
|---|---|---|---|---|
| $d_1$ | (0.2, 0.5) | (0, 0.8) | (0.6, 0.5) | (0.6, 0.3) |
| $d_2$ | (0.4, 0.3) | (0.8, 1) | (0.2, 0.3) | (0.7, 0.6) |
| $d_3$ | (0.9, 0.6) | (0.1, 1) | (0.8, 0.5) | (0.5, 0.8) |
| $d_4$ | (0.7, 0.8) | (0.7, 0.1) | (0, 1) | (0.8, 0.5) |
| $d_5$ | (0.0, 0.8) | (0, 1) | (0, 0.9) | (0, 0.9) |

In matrix form a Z-numbers soft set $(\tilde{\tilde{F}}, A)$ in the above Example can be expressed as follows:

$$\tilde{\tilde{F}}_A = \begin{pmatrix} (0.2, 0.5) & (0.0, 0.8) & (0.6, 0.5) & (0.6, 0.3) \\ (0.4, 0.3) & (0.8, 0.1) & (0.2, 0.3) & (0.7, 0.6) \\ (0.9, 0.6) & (0.1, 1) & (0.8, 0.5) & (0.5, 0.8) \\ (0.7, 0.8) & (0.7, 0.1) & (0.0, 1.0) & (0.8, 0.5) \\ (0.0, 0.8) & (0.0, 1.0) & (0.0, 0.9) & (0.0, 0.9) \end{pmatrix}$$

where the "ith" column vector represents $\tilde{\tilde{F}}(e_i)$, the *ith* row vector represents $d_i$.

If two Z-numbers with the same level of fuzzy restriction, the higher level of the reliability, the larger the Z-numbers would be. First, let us give the related concept on the subset($\subseteq$), equal, intersection($\cap$), union($\cup$), and complement of Z-numbers set, which are defined as follows:

1. $Z_A \subseteq Z_B$, when $\mu_A(x) \leq \mu_B(x)$ and $r_A(x) \leq r_B(x)$;

2. $Z_A = Z_B$, when $\mu_A(x) = \mu_B(x)$ and $r_A(x) = r_B(x)$; or when $Z_A \subseteq Z_B$ and $Z_B \subseteq Z_A$;

3. $Z_A^c = (x, \mu_A^c(x), r_A^c(x)) = (x, 1 - \mu_A(x), 1 - r_A(x))$;;

4. $Z_A \cap Z_B = (x, \mu_{A \cap B}(x), r_{A \cap B}(x)) = (x, \mu_A(x)^* \mu_B(x), r_A(x)^* r_B(x))$;

5. $Z_A \cup Z_B = (x, \mu_{A \cup B}(x), r_{A \cup B}(x)) = (x, \mu_A(x) \circ \mu_B(x), r_A(x) \circ r_B(x))$;

If take standard max and min operations, $\mu_{A \cap B}(x) = \min\{\mu_A(x), \mu_B(x)\}$, $r_{A \cap B}(x) = \min\{r_A(x), r_B(x)\}$. If take $t-norm$ and $t-conorm$ operations, $\mu_A(x)^* \mu_B(x) = \mu_A(x) \cdot \mu_B(x)$, $\mu_A(x) \circ \mu_B(x) = \mu_A(x) + \mu_B(x) - \mu_A(x) \cdot \mu_B(x)$.

In the rest of this paper, we will take $t-norm$ and $t-conorm$ operations to consider the general case. Next, we give the definition on the subset, equal, intersection, union, and complement of Z-numbers soft set.

Definition: Z-numbers Soft Subset. Let $\tilde{\tilde{F}}_A$ and $\tilde{\tilde{G}}_B$ are two Z-numbers soft sets over $(U, E)$. Now $\tilde{\tilde{G}}_B$ is said to be a fuzzy Z-numbers soft subset of $\tilde{\tilde{F}}_A$ if and only if: (1) $B$ is a fuzzy subset of $A$; (2) $G(e)$ is also a fuzzy subset of $F(e)$, $\forall e \in E$.

Consider a Z-numbers soft set $\tilde{\tilde{F}}_A$ over $(U, E)$ given in above Example. Let $\tilde{\tilde{G}}_B$ is another Z-numbers soft set over $(U, E)$ defined as follows:

$\tilde{\tilde{G}}(e_1) = \{(d_1, 0.1, 0.4), (d_2, 0.3, 0.3), (d_3, 0.8, 0.1), (d_4, 0.7, 0.1), (d_5, 0, 1)\}$

$\tilde{\tilde{G}}(e_2) = \{(d_1, 0, 1), (d_2, 0.5, 0.8), (d_3, 0.1, 0.8), (d_4, 0.6, 0.9), (d_5, 0, 1)\}, \tilde{\tilde{G}}(e_4) = \{(d_1, 0.5, 0.2), (d_2, 0.2, 0.5), (d_3, 0.7, 0.1), (d_4, 0, 1), (d_5, 0, 1)\}.$

Then, $\tilde{\tilde{G}}_B$ is a subset of a Z-numbers soft set $\tilde{\tilde{F}}_A$.

Definition: Equality of two Z-numbers soft sets. Let $\tilde{\tilde{F}}_A$ and $\tilde{\tilde{G}}_B$ are two Z-numbers soft sets over $(U, E)$. $\tilde{\tilde{F}}_A$ and $\tilde{\tilde{G}}_B$ are said to be Z-numbers soft equal if and only if $\tilde{\tilde{G}}_B$ is a fuzzy Z-numbers soft subset of $\tilde{\tilde{F}}_A$, and $\tilde{\tilde{F}}_A$ is also a fuzzy Z-numbers soft subset of $\tilde{\tilde{G}}_B$, which can be denoted as $\tilde{\tilde{F}}_A = \tilde{\tilde{G}}_B$.

**Table 2. Complement of $(\tilde{\tilde{F}}, A)$.**

| $U$ | not $e_1$ | not $e_2$ | not $e_3$ | not $e_5$ |
|------|-----------|-----------|-----------|-----------|
| $d_1$ | (0.8, 0.5) | (1.0, 0.2) | (0.4, 0.5) | (0.4, 0.7) |
| $d_2$ | (0.6, 0.7) | (0.2, 0) | (0.8, 0.7) | (0.3, 0.4) |
| $d_3$ | (0.1, 0.5) | (0.9, 0) | (0.2, 0.5) | (0.5, 0.2) |
| $d_4$ | (0.3, 0.2) | (0.3, 0.9) | (1, 0) | (0.2, 0.5) |
| $d_5$ | (1.0, 0.2) | (1, 0) | (1, 0.1) | (1, 0.1) |

Definition: Complement of a Z-numbers soft set. Let $\tilde{\tilde{F}}_A$ is a Z-numbers soft set over $(U, E)$. Then the complement of $\tilde{\tilde{F}}_A$ is denoted $\tilde{\tilde{F}}_A^C$, where, $\tilde{\tilde{F}}_A^C = \{(x_i, (x_i \text{ not satisfies parameter } e_j, 1 - \mu(x_i)_{e_j}, 1 - \gamma(x_i)_{e_j})) : \forall x \in U, \forall e \in E\}$.

In table form, complement of a Z-numbers soft set $\tilde{\tilde{F}}_A$ in the above Example can be expressed as Table 2.

Definition: Union of two Z-numbers soft sets. Let $\tilde{\tilde{F}}_A$ and $\tilde{\tilde{G}}_B$ are two Z-numbers soft sets over $(U, E)$. The union of $\tilde{\tilde{F}}_A$ and $\tilde{\tilde{G}}_B$ is denoted by $\tilde{\tilde{H}}_C = \tilde{\tilde{F}}_A \cup \tilde{\tilde{G}}_B$, where $C = A \cup B$, and

$$\tilde{\tilde{H}}_C(x) = (x, \mu_C(x), r_C(x)) = \begin{cases} (x, \mu_A(x), r_A(x)), & \text{if } A \cup B = A \\ (x, \mu_B(x), r_B(x)), & \text{if } A \cup B = B \\ (x, \mu_A(x) \circ \mu_B(x), r_A(x) \circ r_B(x)), & \text{if } A \cup B = C \end{cases}$$

Definition: Intersection of two Z-numbers soft sets. Let $\tilde{\tilde{F}}_A$ and $\tilde{\tilde{G}}_B$ are two Z-numbers soft sets over $(U, E)$. The intersection of $\tilde{\tilde{F}}_A$ and $\tilde{\tilde{G}}_B$ is denoted by $\tilde{\tilde{H}}_C = \tilde{\tilde{F}}_A \cap \tilde{\tilde{G}}_B$, where $C = A \cap B$, and $\tilde{\tilde{H}}_C(x) = (x, \mu_C(x), r_C(x)) = (x, \mu_A(x) * \mu_B(x), r_A(x) * r_B(x))$.

Let us consider two Z-numbers soft sets $\tilde{\tilde{F}}_A$ and $\tilde{\tilde{G}}_B$ over $(U, E)$ given in the above example. Then, Union of two Z-numbers soft sets $\tilde{\tilde{H}}_C = \tilde{\tilde{F}}_A \cup \tilde{\tilde{G}}_B$ and Intersection of two Z-numbers soft sets $\tilde{\tilde{H}}_C = \tilde{\tilde{F}}_A \cap \tilde{\tilde{G}}_B$ are as Tables 3 and 4.

Definition: Null Z-numbers soft set. Let $\tilde{\tilde{F}}_A$ is a Null Z-numbers soft set over $(U, E)$, when $\tilde{\tilde{F}}(e) = (0, 1)$ for any $e \in E$.

Definition: Absolute Z-numbers soft set. Let $\tilde{\tilde{F}}_A$ is a absolute Z-numbers soft set over $(U, E)$, when $\tilde{\tilde{F}}(e) = (1, 1)$ for any $e \in E$.

**Table 3. $\tilde{\tilde{F}}_A \cup \tilde{\tilde{G}}_B$.**

| $U$ | $e_1$ | $e_2$ | $e_3$ | $e_4$ | $e_5$ |
|------|-------|-------|-------|-------|-------|
| $d_1$ | (0.28, 0.7) | (0, 1) | (0.6, 0.5) | (0.5, 0.2) | (0.6, 0.3) |
| $d_2$ | (0.58, 0.51) | (0.9, 1) | (0.2, 0.3) | (0.2, 0.5) | (0.7, 0.6) |
| $d_3$ | (0.98, 0.64) | (0.18, 1) | (0.8, 0.5) | (0.7, 0.1) | (0.5, 0.8) |
| $d_4$ | (0.91, 0.82) | (0.88, 0.91) | (0, 1) | (0, 1) | (0.8, 0.5) |
| $d_5$ | (0.0, 1) | (0, 1) | (0, 0.9) | (0, 1) | (0, 0.9) |

**Table 4.** $\tilde{\tilde{F}}_A \cap \tilde{\tilde{G}}_B$.

| $U$ | $e_1$ | $e_2$ |
|:---:|:---:|:---:|
| $d_1$ | (0.02, 0.2) | (0, 0.08) |
| $d_2$ | (0.12, 0.09) | (0.4, 0.08) |
| $d_3$ | (0.72, 0.06) | (0.01, 0.8) |
| $d_4$ | (0.49, 0.08) | (0.42, 0.09) |
| $d_5$ | (0.0, 0.08) | (0, 1) |

Let $\tilde{\tilde{F}}_A$ is a Z-numbers soft set over $(U, E)$, then the following holds: (1) $\tilde{\tilde{F}}_A$ is a Z-numbers soft subset of $\tilde{\tilde{F}}_A \cup \tilde{\tilde{F}}_A$. (2) $\tilde{\tilde{F}}_A \cap \tilde{\tilde{F}}_A$ is a Z-numbers soft subset of $\tilde{\tilde{F}}_A$. (3) $\tilde{\tilde{F}}_A \cap \emptyset = \emptyset$. (4) $\tilde{\tilde{F}}_A \cup \emptyset = \tilde{\tilde{F}}_A$. (5) $\tilde{\tilde{F}}_A \cap \tilde{\tilde{A}}' = \tilde{\tilde{F}}_A$. (6) $\tilde{\tilde{F}}_A \cup \tilde{\tilde{A}} = \tilde{\tilde{A}}$.

Let $\tilde{\tilde{F}}_A$, $\tilde{\tilde{G}}_B$, and $\tilde{\tilde{H}}_C$ are any three Z-numbers soft sets over $(U, E)$, then the following holds: (1) $\tilde{\tilde{F}}_A \cup \tilde{\tilde{G}}_B = \tilde{\tilde{G}}_B \cup \tilde{\tilde{F}}_A$; (2) $\tilde{\tilde{F}}_A \cap \tilde{\tilde{G}}_B = \tilde{\tilde{G}}_B \cap \tilde{\tilde{F}}_A$; (3) $\tilde{\tilde{F}}_A \cup (\tilde{\tilde{G}}_B \cup \tilde{\tilde{H}}_C) = (\tilde{\tilde{F}}_A \cup \tilde{\tilde{G}}_B) \cup \tilde{\tilde{H}}_C$; (4) $\tilde{\tilde{F}}_A \cap (\tilde{\tilde{G}}_B \cap \tilde{\tilde{H}}_C) = (\tilde{\tilde{F}}_A \cap \tilde{\tilde{G}}_B) \cap \tilde{\tilde{H}}_C$.

Obliviously, the operational laws of Z-numbers mentioned above are generalized rules of fuzzy set. But it is that the operational laws of Z-numbers soft set mentioned above are not rules of generalized fuzzy soft set.

## 4. A model based on similarity of Z-numbers soft sets

The majority of the decision-making process assume that the decision maker's cognition for all aspects of a problem is the same. However, this is not quite true because of decision maker's inadequate experience, lack of knowledge, different risk preferences and so on. Therefore, it is quite essential to recognize the impact of the decision maker's cognition on the reliability of the information provided. We now develop an approach to solve this kind problem considering the reliability of fuzziness of problem parameters. For given objects(diseases) with certain attributes(symptoms), the Z-numbers soft set as a novel type of soft set can describe the uncertainty that not all the objects satisfy all attributes, but can show details cognitive information of one object satisfying the attribute. Because of the complexity of the real medical diagnosis, using only one aspect of information to describe uncertain events is fully difficult. A new decision-making method with new perspective is presented based on Z-numbers soft set to solve such practical decision-making problems in this section.

We set out the problem in the context of initial medical treatment. Moreover, in the decision framework proposed in this paper, we use technique of similarity measure between two Z-numbers soft sets to estimate the biggest possibility that an ill person is suffering from some certain diseases. Therefore, we first give the similarity on Z-numbers soft sets.

### Similarity on Z-numbers soft sets

First, it is necessary to know whether two patterns or images or alternative are identical or approximately or at least to what degree they are identical. Similarity measure have extensive application in these areas such as pattern (disease) recognition, image processing, region extraction and so on. Some researchers have studied the problem of similarity measure between soft sets, fuzzy soft sets, intuitionistic fuzzy soft sets, interval-valued fuzzy soft sets and so on. Considering the reliability of the information involved in the process, it is required

to compare two Z-numbers soft sets in consequence. Hence, in this section a measure of similarity between two ZnSS has been given, which can solve the calculation on similarities.

Let $U$ refers to the initial non-empty universe of objects, cases, selections and so on, and $E$ is a universal set of parameters, $U = \{x_1, x_2, \cdots, x_m\}$, $E = \{e_1, e_2, \cdots, e_n\}$. Let $\tilde{\tilde{F}}_A$ and $\tilde{\tilde{G}}_B$ be two ZnSS over the parameterized universe $(U, E)$, where $\tilde{\tilde{F}}_A = (\tilde{\tilde{F}}(e_j)) = (x_i, (\mu_A(x_i)_{e_j} \gamma_A(x_i)_{e_j}))$, $\tilde{\tilde{G}}_B = (\tilde{\tilde{G}}(e_j)) = (x_i(\mu_B(x_i)_{e_j}, \gamma_B(x_i)_{e_j}))$, $i = 1, \ldots m, j = 1, \ldots, n$. Then the similarity between $\tilde{\tilde{F}}_A$ and $\tilde{\tilde{G}}_B$ is denoted by $M(\tilde{\tilde{F}}_A, \tilde{\tilde{G}}_B)$, which is expressed as Eq (4):

$$
\begin{aligned}
S(\tilde{\tilde{F}}_A, \tilde{\tilde{G}}_B) &= \max\{M_j(\tilde{\tilde{F}}_A, \tilde{\tilde{G}}_B)\} \\
&= \max\{M_j(\mu_A(x_i), \mu_B(x_i)) \times M_j(\gamma_A(x_i), \gamma_B(x_i))\}
\end{aligned}
\tag{4}
$$

where $M_j(\mu_A(x_i), \mu_B(x_i)) = 1 - \dfrac{\sum_{i=1}^m |\mu_A(x_i)_{e_j} - \mu_B(x_i)_{e_j}|}{\sum_{i=1}^m (\mu_A(x_i)_{e_j} + \mu_B(x_i)_{e_j})}$

and $M_j(\gamma_A(x_i), \gamma_B(x_i)) = 1 - \dfrac{\sum_{i=1}^m |\gamma_A(x_i)_{e_j} - \gamma_B(x_i)_{e_j}|}{\sum_{i=1}^m (\gamma_A(x_i)_{e_j} + \gamma_B(x_i)_{e_j})}$

Consider the following two ZnSS by example:

$$
\tilde{\tilde{F}}_A = \begin{pmatrix}
(0.2, 0.6) & (0.1, 0.8) & (0.2, 0.4) \\
(0.5, 0.8) & (0.2, 0.4) & (0.4, 0.8) \\
(0.9, 0.4) & (0.6, 0.4) & (0.7, 0.6) \\
(1.0, 0.6) & (0.5, 0.6) & (0.9, 0.4)
\end{pmatrix}
$$

$$
\tilde{\tilde{G}}_B = \begin{pmatrix}
(0.4, 0.5) & (0.6, 0.7) & (0.4, 0.9) \\
(0.3, 0.5) & (0.5, 0.6) & (0.3, 0.6) \\
(0.2, 0.6) & (0.2, 0.5) & (0.2, 0.6) \\
(0.9, 0.4) & (0.1, 0.3) & (0.1, 0.5)
\end{pmatrix}
$$

where $U = \{x_1, x_2, x_3, x_4\}$, $E = \{e_1, e_2, e_3\}$. Then,

$$
\begin{aligned}
M_1(\tilde{\tilde{F}}_A, \tilde{\tilde{G}}_B) &= M_1(\mu_A(x_i), \mu_B(x_i)) \times M_j(\gamma_A(x_i), \gamma_B(x_i)) \\
&= \left(1 - \frac{\sum_{i=1}^m |\mu_A(x_i)_{e_1} - \mu_B(x_i)_{e_1}|}{\sum_{i=1}^m (\mu_A(x_i)_{e_1} + \mu_B(x_i)_{e_1})}\right) \times \left(1 - \frac{\sum_{i=1}^m |\gamma_A(x_i)_{e_1} - \gamma_B(x_i)_{e_1}|}{\sum_{i=1}^m (\gamma_A(x_i)_{e_1} + \gamma_B(x_i)_{e_1})}\right) \\
&= \left(1 - \frac{\sum_{i=1}^m |\mu_A(x_i)_{e_j} - \mu_B(x_i)_{e_j}|}{\sum_{i=1}^m (\mu_A(x_i)_{e_j} + \mu_B(x_i)_{e_j})}\right) \times \left(1 - \frac{\sum_{i=1}^m |\gamma_A(x_i)_{e_j} - \gamma_B(x_i)_{e_j}|}{\sum_{i=1}^m (\gamma_A(x_i)_{e_j} + \gamma_B(x_i)_{e_j})}\right) \\
&= \left(1 - \frac{0.2 + +0.2 + 0.7 + 0.1}{0.6 + 0.8 + 1.1 + 1.9}\right) \times \left(1 - \frac{0.1 + 0.3 + 0.2 + 0.2}{1.1 + 1.3 + 0.6 + 1.0}\right) \approx 0.584
\end{aligned}
$$

It is easy to get $M_2(\tilde{\tilde{F}}_A, \tilde{\tilde{G}}_B) \approx 0.358$, $M_3(\tilde{\tilde{F}}_A, \tilde{\tilde{G}}_B) \approx 0.416$.

Hence, the similarity between the two ZnSS over the parameterized universe $(U, E)$ will be $S(\tilde{\tilde{F}}_A, \tilde{\tilde{G}}_B) = \max\{M_1(\tilde{\tilde{F}}_A, \tilde{\tilde{G}}_B), M_2(\tilde{\tilde{F}}_A, \tilde{\tilde{G}}_B), M_3(\tilde{\tilde{F}}_A, \tilde{\tilde{G}}_B)\} = 0.584$.

Let $\tilde{\tilde{F}}_A$ and $\tilde{\tilde{G}}_B$ are two Z-numbers soft set over $(U, E)$, then the following holds:

1. $0 \leq S(\tilde{\tilde{F}}_A, \tilde{\tilde{G}}_B) \leq 1$.

2. $S(\tilde{\tilde{F}}_A, \tilde{\tilde{G}}_B) = S(\tilde{\tilde{G}}_B, \tilde{\tilde{F}}_A)$.

3. $if \tilde{F}_A = \tilde{G}_B, then S(\tilde{F}_A, \tilde{\tilde{G}}_B) = 1$.

4. $if \tilde{\tilde{F}}_A \subseteq \tilde{\tilde{G}}_B \subseteq \tilde{\tilde{H}}_C, then S(\tilde{\tilde{F}}_A, \tilde{H}_C) \leq S(\tilde{\tilde{G}}_B, \tilde{H}_C)$.

5. $if \tilde{\tilde{F}}_A is a crisp soft set, then S(\tilde{\tilde{F}}_A, \tilde{F}_A^C) = 0$.

Proof. Let $\Delta\mu_{FG} = |\mu_A(x_i)_{e_j} - \mu_B(x_i)_{e_j}|$ and $\Delta\gamma_{FG} = |\gamma_A(x_i)_{e_j} - \gamma_B(x_i)_{e_j}|$ for short.

1. From the definition of ZnSSs, we can have $\mu_A(x_i)_{e_j}, \mu_B(x_i)_{e_j}, \gamma_A(x_i)_{e_j}), \gamma_B(x_i)_{e_j} \in [0, 1]$. Then $\min\{\Delta\mu_{FG}, \Delta\gamma_{FG}\} \in [0, 1]$ and $\max\{\Delta\mu_{FG}, \Delta\gamma_{FG}\} \in [0, 1]$.
   We can also obtain that $1 - \min\{\Delta\mu_{FG} \Delta\gamma_{FG}\} \leq 1 + \max\{\Delta\mu_{FG} \Delta\gamma_{FG}\}$, $1 - \min\{\Delta\mu_{FG} \Delta\gamma_{FG}\} \in [0, 1]$, $1 + \max\{\Delta\mu_{FG} \Delta\gamma_{FG}\} \in [1, 2]$. Therefore, $0 \leq S(\tilde{\tilde{F}}_A, \tilde{\tilde{G}}_B) \leq 1$ can be easily obtained.

2. Since $\Delta\mu_{FG} = \Delta\mu_{GF}$ and $\Delta\gamma_{FG} = \Delta\gamma_{GF}$, it is obvious that $S(\tilde{\tilde{F}}_A, \tilde{\tilde{G}}_B) = S(\tilde{\tilde{G}}_B, \tilde{\tilde{F}}_A)$.

3. If $\tilde{\tilde{F}}_A = \tilde{\tilde{G}}_B$, we have $\mu_A(x_i)_{e_j} = \mu_B(x_i)_{e_j}, \gamma_A(x_i)_{e_j} = \gamma_B(x_i)_{e_j}$ for each $x_i \in U$, Then we can get $S(\tilde{\tilde{F}}_A, \tilde{\tilde{G}}_B) = 1$.

4. When $\tilde{\tilde{F}}_A \subseteq \tilde{\tilde{G}}_B \subseteq \tilde{\tilde{H}}_C$, it has $\mu_A(x_i)_{e_j} \leq \mu_B(x_i)_{e_j} \leq \mu_C(x_i)_{e_j}$, and $\gamma_A(x_i)_{e_j} \geq \gamma_B(x_i)_{e_j} \geq \gamma_C(x_i)_{e_j}$ for each $x_i \in U$. Then we have $|\mu_A(x_i)_{e_j} - \mu_C(x_i)_{e_j}| \geq |\mu_A(x_i)_{e_j} - \mu_B(x_i)_{e_j}|$ and $|\gamma_A(x_i)_{e_j} - \gamma_C(x_i)_{e_j}| \geq |\gamma_A(x_i)_{e_j} - \gamma_B(x_i)_{e_j}|$ for each $x_i \in U$. Thus,

$$\min\{|\mu_A(x_i)_{e_j} - \mu_C(x_i)_{e_j}|, |\gamma_A(x_i)_{e_j} - \gamma_C(x_i)_{e_j}|\} \geq \min\{|\mu_A(x_i)_{e_j} - \mu_B(x_i)_{e_j}|,$$
$$|\gamma_A(x_i)_{e_j} - \gamma_B(x_i)_{e_j}|\} \cdot \max\{|\mu_A(x_i)_{e_j} - \mu_C(x_i)_{e_j}|, |\gamma_A(x_i)_{e_j} - \gamma_C(x_i)_{e_j}|\} \geq$$
$$\max\{|\mu_A(x_i)_{e_j} - \mu_B(x_i)_{e_j}|, |\gamma_A(x_i)_{e_j} - \gamma_B(x_i)_{e_j}|\}.$$

From operations on Z-numbers soft set, we can get $S(\tilde{\tilde{F}}_A, \tilde{H}_C) \leq S(\tilde{\tilde{G}}_B, \tilde{H}_C)$ easily.

5. If $\tilde{\tilde{F}}_A$ is a crisp soft set, we have $\tilde{\tilde{F}}_A = \{(\mu, (1, 0))\}$ or $\tilde{\tilde{F}}_A = \{(\mu, (0, 1))\}$. Then, the complement set of $\tilde{\tilde{F}}_A$ an be calculated as $\tilde{\tilde{F}}_A^C = \{(\mu, (0, 1))\}$ or $\tilde{\tilde{F}}_A^C = \{(\mu, (1, 9))\}$ respectively. Thus, $S(\tilde{\tilde{F}}_A, \tilde{\tilde{F}}_A^C) = 0$.

## Decision framework

In this section, we introduce the following decision framework to estimate the biggest possibility that an ill person is suffering from some certain disease in medical diagnosis under ZnSS environment considering the reliability of the information for the different observers' cognition. Supposed that a patient with a lot of nonspecific symptoms needs to figure out where is the most likely disease so as to help him register effectively to the appropriate section. Hence, we will try to discuss approaches to estimate the biggest possibility-disease recognition from a multi-observer data for an ill person having uncertain symptoms in this paper. Fig 1 shows decision framework in medical diagnosis based on Z-numbers soft set.

The decision framework involves the following steps:

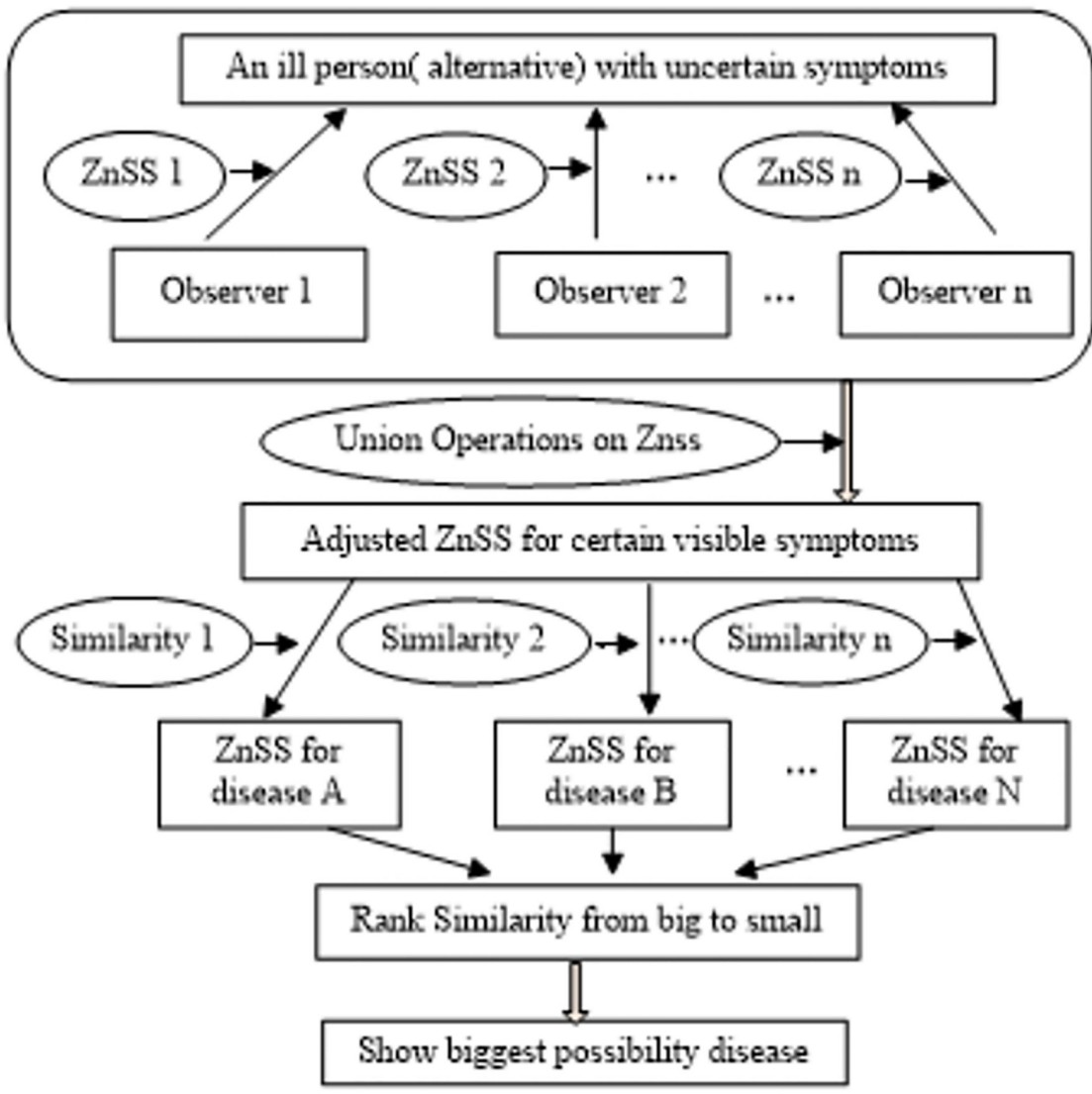

**Fig 1. Decision framework based on ZnSS.**

Step 1: Construct the model Z-numbers soft sets for different diseases. In this paper it is given that the model Z-numbers soft sets for different diseases denoted by $\tilde{\tilde{D}}_A, \tilde{\tilde{D}}_B$...

Step 2: Give judgment information of different observers on the ill person. In the majority situation, the judgment information is basically similar to "fever will surely be very high", "cough will high probability be severe", etc. And a different set of symptoms is likely to be selected by different observers. The concept of Z-numbers soft set presented in this paper has significant potential in the describing of the uncertainty of the human knowledge because it consists of restraint and reliability of the measured value in details.

Hence, here we assumed that the observers provide their judgment information on the alternatives with regard to attributes by Z-numbers soft sets. Thus, the observers $O_k(k = 1, 2, \cdots, l)$ provides their judgments information for attributes $e_j(j = 1, 2, \cdots, n)$, which can be expressed by Z-numbers soft sets $\tilde{\tilde{P}}_{O_k}(e_j)$.

Step 3: Calculate the union operations on all Z-numbers soft sets $\tilde{\tilde{P}}_{O_k}(e_j)$ of different observers, which is denoted by $\tilde{\tilde{H}}_C = \cup \tilde{\tilde{P}}_{O_k}(e_j)$, $k = 1, 2, \ldots, l$ and $j = 1, 2, \cdots, n$. Then by the repeated use of $\tilde{\tilde{P}}_{O_1} \cup (\tilde{\tilde{P}}_{O_2} \cup \tilde{\tilde{P}}_{O_3}) = (\tilde{\tilde{P}}_{O_1} \cup \tilde{\tilde{P}}_{O_2}) \cup \tilde{\tilde{P}}_{O_3}$ and

$$
\tilde{\tilde{P}}_{O_1} \cup \tilde{\tilde{P}}_{O_2} = 
\begin{cases}
(x, \mu_{O_1}(x), r_{O_1}(x)), & \text{if } O_1 \cup O_2 = O_1 \\
(x, \mu_{O_2}(x), r_{O_2}(x)), & \text{if } O_1 \cup O_2 = O_2 \\
(x, \mu_{O_1}(x) \circ \mu_{O_2}(x), r_{O_1}(x) \circ r_{O_2}(x)), & \text{if } O_1 \cup O_2 \neq O_1 \neq O_2
\end{cases}
$$

The final result is that we can get the adjusted Z-numbers soft set $\tilde{\tilde{H}}_C$ on the visible symptoms of the ill person by integrated observation and judgment from the observers $O_k(k = 1, 2, \cdots, l)$.

Step 4: Calculate and obtain the similarity between the adjusted Z-numbers soft set $\tilde{\tilde{H}}_C$ and the model Z-numbers soft sets for different diseases denoted by $\tilde{\tilde{D}}_A, \tilde{\tilde{D}}_B \ldots$ Here, it need to apply the method of similarity measure on different ZnSS. The similarity between the adjusted Z-numbers soft set $\tilde{\tilde{H}}_C$ and the model Z-numbers soft sets for different diseases are denoted by $S(\tilde{\tilde{H}}_C, \tilde{\tilde{D}}_A), S(\tilde{\tilde{H}}_C, \tilde{\tilde{D}}_B) \ldots$

Step 5: Rank the similarity $S(\tilde{\tilde{H}}_C, \tilde{\tilde{D}}_A), S(\tilde{\tilde{H}}_C, \tilde{\tilde{D}}_B) \ldots$ from big to small. Identify the most likely disease. Choose the biggest score of similarities, the corresponding disease is the most likely disease.

## 5. Case study

In this paper there is only a simple example only with two diseases under consideration (influenza and COVID-19) to show the possibility of using this approach-based Z-numbers soft set, not a medical diagnosis in a real-world scenario, which is similar to the existing studies [32, 40]. But the method can be generalized in practice. On the one hand, there are not only two choices in reality to estimate for preliminary diagnosis of disease which could be improved by clinical results. In that case, the ranking results of the evaluation can help patients make the choice of registration ranking. On the other hand, the work in this paper can also help develop a platform for primary diagnosis.

### Illustrative example

In the following example, we will try to estimate the biggest possibility that an ill person with some fuzzy symptoms is suffering from some kind of disease in medical diagnosis. For example, an ill person has some symptoms in the outbreak of COVID-19. It is quite essential to preliminary estimate by some observers or some online diagnose that the disease is more likely to be a influenza or a COVID-19 for shortage of offline outpatient service. In this paper the prim symptoms under consideration include fever ($e_1$), cough with chest congestion ($e_2$), runny nose ($e_3$), body ache ($e_4$), headache ($e_5$), breathing trouble ($e_6$), diarrhea ($e_7$), sore throat ($e_8$). To be sure the observers often provide the symptom information roughly the same as "fever will surely be very high". Hence, the judgment of the observers on the symptoms can be express by different Z-numbers soft set, in which we use converted [0, 1] value of degree that alternative $x_i$ have parameter $e_j$ and the degree of reliability. We let the universal set only contain two elements "yes" and "no", i.e., $U = \{y, n\}$, and $E = \{e_1, e_2, e_3, e_4, e_5, e_6, e_7, e_8\}$.

In this example it is given that the model Z-numbers soft sets for a influenza $\tilde{\tilde{F}}_A$ in Table 5 and a COVID-19 $\tilde{\tilde{G}}_B$ in Table 6. Now there are three related observers provide the symptom

**Table 5. A ZnSS model for influenza $\tilde{\tilde{F}}_A$.**

| $\tilde{\tilde{F}}_A$ | $e_1$ | $e_2$ | $e_3$ | $e_4$ | $e_5$ | $e_6$ | $e_7$ | $e_8$ |
|---|---|---|---|---|---|---|---|---|
| y | (0.5, 0.8) | (0.5, 0.8) | (0.9, 0.9) | (0.2, 0.8) | (0.8, 0.9) | (0.2, 0.8) | (0.2, 0.9) | (0.8, 0.9) |
| n | (0.5, 0.8) | (0.5, 0.8) | (0.1, 0.9) | (0.7, 0.8) | (0.2, 0.8) | (0.8, 0.8) | (0.8, 0.9) | (0.2, 0.8) |

**Table 6. A ZnSS model for COVID-19 $\tilde{\tilde{G}}_B$.**

| $\tilde{\tilde{F}}_A$ | $e_1$ | $e_2$ | $e_3$ | $e_4$ | $e_5$ | $e_6$ | $e_7$ | $e_8$ |
|---|---|---|---|---|---|---|---|---|
| y | (0.9, 0.9) | (0.8, 0.8) | (0.1, 0.9) | (0.5, 0.9) | (0.5, 0.9) | (0.8, 0.9) | (0.5, 0.9) | (0.6, 0.9) |
| n | (0.1, 0.9) | (0.1, 0.7) | (0.8, 0.9) | (0.5, 0.9) | (0.5, 0.9) | (0.1, 0.9) | (0.5, 0.9) | (0.4, 0.9) |

information, which can be constructed by Z-numbers soft sets $\tilde{\tilde{P}}_{O_1}$ in Table 7, $\tilde{\tilde{P}}_{O_2}$ in Table 8, and $\tilde{\tilde{P}}_{O_3}$ in Table 9.

## Calculation process

Step 1: Construct and input the model Z-numbers soft sets $\tilde{\tilde{F}}_A$ and $\tilde{\tilde{G}}_B$ for a influenza and a COVID-19 in Tables 5 and 6.

Step 2: Obtain the symptom information from the 3 observers $\tilde{\tilde{O}}_1$, $\tilde{\tilde{O}}_2$ and $\tilde{\tilde{O}}_3$, then construct and input the corresponding Z-numbers soft sets $\tilde{\tilde{P}}_{O_1}$ in Table 7, $\tilde{\tilde{P}}_{O_2}$ in Table 8, and $\tilde{\tilde{P}}_{O_3}$ in Table 9.

**Table 7. A ZnSS for the ill person from observer 1 $\tilde{\tilde{P}}_{O_1}$.**

| $\tilde{\tilde{F}}_A$ | $e_1$ | $e_2$ | $e_3$ | $e_4$ | $e_5$ | $e_6$ | $e_7$ | $e_8$ |
|---|---|---|---|---|---|---|---|---|
| y | (0.5, 0.5) | (0.4, 0.8) | (0.8, 0.7) | (0.2, 0.8) | (0.2, 0.8) | (0.5, 0.5) | (0.5, 0.5) | (0.6, 0.5) |
| n | (0.5, 0.8) | (0.5, 0.8) | (0, 1) | (0.7, 0.7) | (0.6, 0.7) | (0.4, 0.5) | (0.4, 0.5) | (0.2, 0.7) |

**Table 8. A ZnSS for the ill person from observer 2 $\tilde{\tilde{P}}_{O_2}$.**

| $\tilde{\tilde{F}}_A$ | $e_1$ | $e_2$ | $e_3$ | $e_4$ | $e_5$ | $e_6$ | $e_7$ | $e_8$ |
|---|---|---|---|---|---|---|---|---|
| y | (0.8, 0.8) | (0.5, 0.5) | (0.5, 0.7) | (0.5, 0.5) | (0.5, 0.5) | (0.4, 0.6) | (0.2, 0.8) | (0.5, 0.6) |
| n | (0.1, 0.5) | (0.5, 0.6) | (0.2, 0.8) | (0.5, 0.5) | (0.5, 0.5) | (0.4, 0.7) | (0.5, 0.5) | (0.3, 0.6) |

**Table 9. A ZnSS for the ill person from observer 3 $\tilde{\tilde{P}}_{O_3}$.**

| $\tilde{\tilde{F}}_A$ | $e_1$ | $e_2$ | $e_3$ | $e_4$ | $e_5$ | $e_6$ | $e_7$ | $e_8$ |
|---|---|---|---|---|---|---|---|---|
| y | (0.6, 0.7) | (0.2, 0.5) | (0.5, 0.5) | (0.4, 0.7) | (0.4, 0.7) | (0.5, 0.6) | (0.4, 0.5) | (0.6, 0.5) |
| n | (0.1, 0.8) | (0.2, 0.5) | (0.3, 0.6) | (0.5, 0.7) | (0.4, 0.7) | (0.4, 0.5) | (0.4, 0.6) | (0.3, 0.5) |

**Table 10. ZnSS $\tilde{\tilde{H}}_C$ for union operations for $\tilde{\tilde{H}}_{\cup C} = \tilde{\tilde{P}}_{O_1} \cup \tilde{\tilde{P}}_{O_2} \cup \tilde{\tilde{P}}_{O_3}$.**

| $\tilde{\tilde{F}}_A$ | $e_1$ | $e_2$ | $e_3$ | $e_4$ | $e_5$ | $e_6$ | $e_7$ | $e_8$ |
|---|---|---|---|---|---|---|---|---|
| y | (0.96, 0.97) | (0.96, 0.95) | (0.95, 0.5) | (0.76, 0.97) | (0.76, 0.97) | (0.75, 0.96) | (0.76, 0.95) | (0.92, 0.9) |
| n | (0.59, 0.98) | (0.8, 0.96) | (0.44, 1) | (0.92, 0.95) | (0.88, 0.95) | (0.78, 0.92) | (0.82, 0.9) | (0.6, 0.94) |

Step 3: Calculate the union operations on all Z-numbers soft sets, $\tilde{\tilde{H}}_{\cup C} = \tilde{\tilde{P}}_{O_1} \cup \tilde{\tilde{P}}_{O_2} \cup \tilde{\tilde{P}}_{O_3}$. Now we get he adjusted Z-numbers soft set on the visible symptoms of the ill person by the integrated observation and judgment from the observers, seen as Table 10.

Step 4: Calculate the similarity between the adjusted Z-numbers soft set $\tilde{\tilde{H}}_C$ and the model Z-numbers soft sets for different diseases the model Z-numbers soft sets $\tilde{\tilde{F}}_A$ and $\tilde{\tilde{G}}_B$ for a influenza and a COVID-19 in Tables 5 and 6.

1.

$$\begin{aligned} S(\tilde{\tilde{H}}_C, \tilde{\tilde{F}}_A) &= \max\{M_j(\tilde{\tilde{H}}_C, \tilde{\tilde{F}}_A)\} \\ &= \max\{M_j(\mu_C(x_i), \mu_A(x_i)) \times M_j(\gamma_C(x_i), \gamma_A(x_i))\} \end{aligned}$$

where $M_1(\tilde{\tilde{H}}_C, \tilde{\tilde{F}}_A) = 0.709, M_2(\tilde{\tilde{H}}_C, \tilde{\tilde{F}}_A) = 0.629, M_3(\tilde{\tilde{H}}_C, \tilde{\tilde{F}}_A) = 0.710,$ $M_4(\tilde{\tilde{H}}_C, \tilde{\tilde{F}}_A) = 0.634, M_5(\tilde{\tilde{H}}_C, \tilde{\tilde{F}}_A) = 0.640, M_6(\tilde{\tilde{H}}_C, \tilde{\tilde{F}}_A) = 0.712, M_7(\tilde{\tilde{H}}_C, \tilde{\tilde{F}}_A) = 0.765,$ $M_8(\tilde{\tilde{H}}_C, \tilde{\tilde{F}}_A) = 0.899,$

2.

$$\begin{aligned} S(\tilde{\tilde{H}}_C, \tilde{\tilde{G}}_B) &= \max\{M_j(\tilde{\tilde{H}}_C, \tilde{\tilde{G}}_B)\} \\ &= \max\{M_j(\mu_C(x_i), \mu_B(x_i)) \times M_j(\gamma_C(x_i), \gamma_B(x_i))\} \end{aligned}$$

where $M_1(\tilde{\tilde{H}}_C, \tilde{\tilde{G}}_B) = 0.753, M_2(\tilde{\tilde{H}}_C, \tilde{\tilde{G}}_B) = 0.803, M_3(\tilde{\tilde{H}}_C, \tilde{\tilde{G}}_B) = 0.402,$ $M_4(\tilde{\tilde{H}}_C, \tilde{\tilde{G}}_B) = 0.722, M_5(\tilde{\tilde{H}}_C, \tilde{\tilde{G}}_B) = 0.571, M_6(\tilde{\tilde{H}}_C, \tilde{\tilde{G}}_B) = 0.684, M_7(\tilde{\tilde{H}}_C, \tilde{\tilde{G}}_B) = 0.765,$ $M_8(\tilde{\tilde{H}}_C, \tilde{\tilde{G}}_B) = 0.760,$

Hence, $S(\tilde{\tilde{H}}_C, \tilde{\tilde{F}}_A) = 0.899, S(\tilde{\tilde{H}}_C, \tilde{\tilde{G}}_B) = 0.803$

Step 5: Identify the most likely disease.

In this example, $S(\tilde{\tilde{H}}_C, \tilde{\tilde{F}}_A)$ is bigger, so the ill person is most likely getting a influenza.

## Result discussion and comparisons

Handing uncertainty in medical diagnose is an open issue. In this subsection, we compare the proposed group medical decision-making model with other existing decision-making approaches.

In the existed approaches, theory of fuzzy soft sets is applied in the field of medical decision making at the earliest. As practical applications, the intuitionistic fuzzy soft sets have been successfully applied into medical system [32], and group decision making method using intuitionistic fuzzy soft sets is of great significance in aiding medical diagnosis [40]. Although the traditional uncertainty mathematical theories have developed greatly and achievements have been widely applied in medical diagnosis fields. There is a challenge is how to measure reliability of the information and how to handle the uncertainty effectively to achieve more accurate

Table 11. Comparison results of ZnSS decision making methods.

| Method | Diagnosis result | Ranking order |
|---|---|---|
| Hamming distance [32] | Influenza | $\mu_3 > \mu_1 = \mu_5 > \mu_2 = \mu_4$ |
| Euclidean distance [32] | Influenza | $\mu_3 > \mu_1 = \mu_5 = \mu_2 = \mu_4$ |
| Non-normalized IFSM [32] | Influenza | $\mu_3 > \mu_2 > \mu_5 = \mu_1 > \mu_4$ |
| Normalized IFSM [32] | Influenza | $\mu_3 > \mu_2 > \mu_5 > \mu_1 > \mu_4$ |
| G-IFSS method [40] | Influenza | $\mu_3 > \mu_5 > \mu_1 > \mu_2 = \mu_4$ |
| Saeed et al. [66] | Influenza | $\mu_3 > \mu_2 > \mu_5 > \mu_1 > \mu_4$ |
| Riaz et al. [67] | Influenza | $\mu_3 > \mu_2 > \mu_1 > \mu_5 > \mu_4$ |
| Zulqarnain, R. M., et al. [68] | Influenza | $\mu_3 > \mu_2 > \mu_1 > \mu_4 > \mu_5$ |

where $\mu_1$: Viral, $\mu_2$: fever Malaria, $\mu_3$: Influenza, $\mu_4$: Gastric ulcer, $\mu_5$: Pneumonia.

decision-making. Diagnosing diseases by uncertainty symptoms is not simple task at all. Hence, an approach based on Z-numbers soft set is proposed in this paper. Our proposed method uses the similar example and datasets, which have been analyzed by four existing approaches in [32]. In the method proposed in [32], the difference is measured between diagnoses of each expert and medical knowledgebase using Hamming and Euclidean distance for a patient. The smaller the difference obtained, the more appropriate the diagnosis appears. In [32] researchers also utilized a confident weight assigning mechanism. The larger the weight of the disease is assigned, the more likely the patient suffers from this kind of disease. The ranking results of these approaches are exhibited in Table 11. The disease the patient most likely suffers from is influenza with the method proposed in [40], which is also exhibited in Table 11. The proposed method is also compared with other existing methods: Saeed et al. [66], Riaz et al. [67] and Zulqarnain, R. M., et al. [68]. The comparison results are listed in Table 11.

We can find that the disease the patient most likely suffers from is Influenza with our proposed method, which is same as described in the previous five approaches. If we use our method to rank, the ranking of the alternatives in proposed method is almost similar as obtained by the fourth approach, which indicates that the new method works well. However, there are some differences existed in the results between the proposed method and other five methods. The reason why the proposed method is desirable is ascribed to its own uniqueness. The proposed method is Znss-based similarity measures. Z-numbers are composed of two parts: one part denotes a restriction on values that can be assumed, and another part is the reliability of the information. The properties are different and the result is reasonably inconsistent. In fact, because medical diagnosis involves significant fuzzy concepts and uncertainties, the reliability of information is very important. If only the first component is considered while the second component is ignored, then the reliability of information may be limited, which can lead to an incorrect result. In addition, the existing approaches that are compared in the paper all pay attention to the weight parameter. While they only take single subjective of objective factor into consideration. The new method considers both to get the t parameter. More useful information is considered in the new method. The ranking result may change with different approaches, but the method proposed in this paper considers more evaluation information.

## 6. Conclusion

The notion of the Z-numbers soft set is proposed in this paper in order to handle medical diagnose problem, in which the uncertainty of events from both the macro-angle and the micro-angle could be presented. For given objects with certain attributes, the soft set theory can be

used to describe the uncertainty that not all the objects satisfy all attributes, while the application of Z-numbers can show detail cognitive information of one object satisfying the attribute. Z-numbers soft set is a combination of Z-numbers theory and soft set theory. The subset, equal, intersection, union and complement operations are also defined on the Z-numbers soft set. The basic properties of the Z-numbers soft sets are also presented. Similarity measure of two Z-numbers soft sets is discussed, and an approach to medical diagnosis has been developed based on similarity measure of Z-numbers soft sets. An illustrative example is showed. This new extension not only provides an addition to existing theories for handling uncertainties especially in handle the reliability of fuzziness of problem parameters.

The advantages of the proposed method are summarized below.

1. The traditional uncertainty mathematical theories have developed greatly and achievements have been widely applied in medical diagnosis. However, the traditional uncertainty mathematical theories have their intrinsic difficulties, which are pointed by Molodtsov [19]. Soft set theory proposed by Molodtsov has been regarded as an effective mathematical tool to deal with uncertainty. The method presented in this paper based fuzzy extensions of soft set theory are presented can express different fuzziness of medical diagnose parameters effectively.

2. Fuzzy numbers have been widely applied in decision making of medical diagnose. However, we found that the reliability of uncertainty symptoms in medical diagnose environments is also important. To solve this situation, Z-numbers are used to model and describe the diagnoses of decision-makers on uncertainty symptoms. Z-numbers combined with the constraint and reliability.

3. When applying Z-numbers, we need an appropriate method for express different fuzziness of medical diagnose parameters and handle the reliability effectively. To address these problems, we combine the soft set and Z-numbers, we propose the notion of the Z-numbers soft set and treat the Z-numbers soft set as a whole, rather than converting the second component, to avoid the loss of symptoms information.

4. Similarity measure have extensive application in the area of disease recognition. Considering the reliability of the information involved in the process, a measure of similarity between two ZnSS has been given in this paper to compare two Z-numbers soft sets in consequence, which can solve the calculation on similarities to help the final diagnose result.

5. In the real the diagnoses of decision-making problems, the proposed method can obtain reasonable and effective results, as demonstrated by comparing the obtained results with those from the existing methods. This method can also be applied to other multi-attribute decision-making problems.

The study in this paper is an interdisciplinary approach towards rapid and efficient medical diagnosis. The approach based on Z-numbers soft set can measure reliability of the information and handle the uncertainty effectively to achieve more accurate decision-making. Although the proposed approach has been demonstrated to be effective through illustrative examples and in-depth discussion, there are still some aspects and potential areas that can be improved in future studies. The difference in importance between two components of a Z-number in Z-number soft set, namely the assessment value and the reliability measure need to be studied further. In this paper, determine the weight of the two components remains an unresolved issue, though the importance of these two components should be different. Second, in the proposed method for compare two Z-numbers soft set using similarity measure without considering the association between parameters. Third, the feasibility and effectiveness of the

method are just verified by numerical examples rather than real professional medical knowledge in this paper. Therefore, the future study directions will include the parameterization reduction of Z-numbers soft sets. It is also desirable to further explore the applications of using the Z-numbers soft sets approach to solve real world specific problems in the process of decision making in medical diagnosis.

## Supporting information

**S1 File. Medical knowledgebase of illustrative example.**
(PDF)

**S2 File. Medical knowledgebase of comparative example.**
(PDF)

## Acknowledgments

The authors greatly appreciate the reviewers' suggestions and the editors' encouragement.

## Author Contributions

**Conceptualization:** Haiyan Zhao.

**Formal analysis:** Zheng Liu.

**Funding acquisition:** Yanhong Wang.

**Writing – original draft:** Haiyan Zhao.

**Writing – review & editing:** Qian Xiao, Yanhong Wang.

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
