## [Decision Letter · Decision Letter 0]

16 Mar 2022

PONE-D-21-34053An approach in medical diagnosis based on Z-numbers soft setPLOS ONE

Dear Dr. Xiao,

Thank you for submitting your manuscript to PLOS ONE. After careful consideration, we feel that it has merit but does not fully meet PLOS ONE’s publication criteria as it currently stands. Therefore, we invite you to submit a revised version of the manuscript that addresses the points raised during the review process.

We look forward to receiving your revised manuscript.

Kind regards,

Ziqiang Zeng, Ph.D.

Academic Editor

PLOS ONE

Journal Requirements:

The works described in this paper are supported by the National Social Science Fund of

China under Grant No. 18CGL0

Liu Zheng is funded by the National Social Science Fund of China under Grant No. 18CGL015. The funders had no role in study design, data collection and analysis, decision to publish, or preparation of the manuscript.

5. Please include a caption for figure 1.

Additional Editor Comments:

Based on the reviewers' comments, this paper has a potential to be further considered if the authors can well address all the issues raised in the review comments. A detailed response to all the reviewers is needed when the authors submit the revised manuscript.

Reviewers' comments:

Reviewer's Responses to Questions

**Comments to the Author**

1. Is the manuscript technically sound, and do the data support the conclusions?

Reviewer #1: Yes

Reviewer #2: Yes

Reviewer #3: Yes

2. Has the statistical analysis been performed appropriately and rigorously? 

Reviewer #1: N/A

Reviewer #2: Yes

Reviewer #3: I Don't Know

3. Have the authors made all data underlying the findings in their manuscript fully available?

Reviewer #1: Yes

Reviewer #2: Yes

Reviewer #3: No

4. Is the manuscript presented in an intelligible fashion and written in standard English?

Reviewer #1: Yes

Reviewer #2: Yes

Reviewer #3: Yes

5. Review Comments to the Author

Reviewer #1: This manuscript proposes a new concept of Z-number soft sets and defines some operations on Z-number soft sets, as well as a similarity measure for two Z-number soft sets. From the general perspective, the paper is interesting, however, some issues should be faced to improve the paper.

There are some suggestions and comments as follows:

1. The presentation quality of articles needs to be improved urgently. Some paragraphs and formats do not read well in this manuscript. There are many sentences in the article with obvious grammatical errors or unclear expressions. For example, there are three grammatical errors in the last sentence of the abstract; in subsection 2.2, the bibliography is cluttered, etc. The author should carefully check and modify.

2. The abstract is loosely written. It is not as informative as expected. A standard abstract must present, without leaving any doubt, the objective and contribution of the paper precisely.

3. Authors should add more descriptions of motivation, both in the abstract and the introduction.

4. A detailed proof should be given for the similarity measurement properties of two Z-number soft sets, rather than a sentence "The results trivially straightforward from definition".

5. The article lacks a comparison with other existing studies.

6. The case in Section 5 is just a simple example, not a medical diagnosis in a real-world scenario, and provides little useful information in the Discussion section.

7. Please explain the similarities and differences between the "ZnSS" in the manuscript and the "Z-set" proposed in the research "Z-set Based Approach to Control System Design".

Reviewer #2: The authors use the Zadeh-fuzzy number in diagnosis in medicine. The paper is solid and uses the soft set in its application

The authors should mention into the paper the extension of Soft Set to HyperSoft Set, i.e.:

In 2018 Smarandache generalized the Soft Set to the HyperSoft Set by transforming the classical uni-argument function F into a multi-argument function:

F. Smarandache: Extension of Soft Set to Hypersoft Set, and then to Plithogenic Hypersoft Set, Neutrosophic Sets and Systems, vol. 22, 2018, pp. 168-170. DOI: 10.5281/zenodo.2159754

http://fs.unm.edu/NSS/ExtensionOfSoftSetToHypersoftSet.pdf

Reviewer #3: The paper proposed an approach to medical diagnosis and used to estimate whether two patterns or images are identical or approximately.In my opinion, this paper contains some interesting results which make a significant and technically sound contribution to the field. However, there are some issues that should be considered in the revised version.

-Abstract should be restated by adding the importance and contribution of the work.

-Introduction section should be completely updated by adding motivation, organization and novelty of the work.

-The author should added the last two year reference related the proposed work and link the proposed work. Please update the reference and citation.

-Authors should use the common symbols which can be easily understandable and readable

-What effect is the use of the proposed model in achieving the objectives of the research? It is suggested that the results of the proposed model are stated in the conclusion with full detail.

-In addition to expressing the superiority of the proposed method, its challenges need to be addressed.

-In the research methodology section, explain why this idea was proposed and what is its superiority over other methods?

-In the related works section, new resources will be used and implicitly referred to the challenges of previous methods.

-It is necessary to state in the related works section, why the previous methods are not responsible for solving the research problem accurately and you need to use to a new method?

-There are too many grammatical and writing errors. The paper needs proofreading.such as:

1. in "Preliminaries"section formulas must be written more clearly.

2.citation of all references must be corrected.

3.delete title "3.1".

4. in page 8., why some sentences have been bolded.

5. The titles of tables(such as Table 3,4) are not true.

6. the title of section 4 is similar to the title of this paper. Change it.

7. Change title 4.3

8. the explanation of the steps of Algorithm are not very clear.

9.after title 5, please insert some words.

-For better understand, send me your code to validate your paper.

-The calculations have not been analyzed

-Conclusion section should be rewritten.

-The author should enrich the references section by adding the recent following references in the paper in this field and also other types of uncertainty:

[] (2018). A proposed model for solving fuzzy linear fractional programming problem: Numerical Point of View. Journal of computational science, 25, 367-375.

[](2019). A novel approach to solve gaussian valued neutrosophic shortest path problems. International Journal of Engineering and Advanced Technology, Volume-8 Issue-3,347-353

[](2020). Data envelopment analysis based on triangular neutrosophic numbers. CAAI transactions on intelligence technology, 5(2), 94-98.

[](2020). A new decision making approach for winning strategy based on muti soft set logic. Journal of Fuzzy Extension and Applications, 1(2), 112-121.

[](2020). Neutrosophic structured element. Expert systems, 37(5), e12542.

[](2021). The multi-fuzzy N-soft set and its applications to decision-making. Neural Computing and Applications, 33(17), 11437-11446.

[](2021). Fuzzy hypersoft sets and its weightage operator for decision making. Journal of Fuzzy Extension and Applications, 2(2), 163-170.

[] (2021). A hybrid decision-making analysis under complex q-rung picture fuzzy Einstein averaging operators. Computational and Applied Mathematics, 40(8), 1-35.

[](2021). A survey on different deﬁnitions of soft points: limitations, comparisons and challenges. Journal of Fuzzy Extension and Applications, 2(4), 333-343.

[](2022). IFP-intuitionistic multi fuzzy N-soft set and its induced IFP-hesitant N-soft set in decision-making. Journal of Ambient Intelligence and Humanized Computing, 1-10.

[] (2021). Combined probabilistic linguistic term set and ELECTRE II method for solving a venture capital project evaluation problem. Economic Research-Ekonomska Istraživanja, 1-23.

[](2022). Complex fermatean fuzzy N-soft sets: a new hybrid model with applications. Journal of Ambient Intelligence and Humanized Computing, 1-34.

6. PLOS authors have the option to publish the peer review history of their article (what does this mean?). If published, this will include your full peer review and any attached files.

Reviewer #1: No

Reviewer #2: No

Reviewer #3: No

---

## [Author Response · Author response to Decision Letter 0]

9 May 2022

Dear editorial board of PLOS ONE,

Thank you for considering the revised version of our manuscript “An approach in medical diagnosis based on Z-numbers soft set”. We appreciate editor and reviewers very much for their positive and constructive comments and suggestions on our manuscript. 

We have thoughtfully taken into account these comments. The explanation of what we have changed in response to the reviewers’ concerns is given point by point in the file named “Response to Reviewer（PONE-D-21-34053）”. We believe that the comments have been highly constructive and very useful to restructure the manuscript. 

We have tried our best to revise our manuscript according to the Journal Requirements. We have updated our submission to use the PLOS LaTeX template to ensure that your manuscript meets PLOS ONE's style requirements. We have added a caption for figure 1. And we have no changes to our Data Availability statement. We have uploaded the Supporting Information file, and we ensure that the file we include separate captions for our supplementary files at the end of our manuscript. We also have corrected the grant numbers in the ‘Funding Information’ section.

We hope that all these changes fulfil the requirements to make the manuscript acceptable for publication in PLOS ONE. We would like to express our great appreciation to you and reviewers for comments on our paper.

Looking forward to hearing from you soon.

Many thanks and best regards.

Sincerely yours,

Qian Xiao on behalf of the authors.

Corresponding author: 

Name: Qian Xiao, 

E-mail: 817589@qq.com

---

## [Decision Letter · Decision Letter 1]

8 Jun 2022

PONE-D-21-34053R1An approach in medical diagnosis based on Z-numbers soft setPLOS ONE

Dear Dr. Xiao,

Thank you for submitting your manuscript to PLOS ONE. After careful consideration, we feel that it has merit but does not fully meet PLOS ONE’s publication criteria as it currently stands. Therefore, we invite you to submit a revised version of the manuscript that addresses the points raised during the review process.

We look forward to receiving your revised manuscript.

Kind regards,

Ziqiang Zeng, Ph.D.

Academic Editor

PLOS ONE

Journal Requirements:

Reviewers' comments:

Reviewer's Responses to Questions

**Comments to the Author**

1. If the authors have adequately addressed your comments raised in a previous round of review and you feel that this manuscript is now acceptable for publication, you may indicate that here to bypass the “Comments to the Author” section, enter your conflict of interest statement in the “Confidential to Editor” section, and submit your "Accept" recommendation.

Reviewer #1: (No Response)

Reviewer #2: All comments have been addressed

Reviewer #3: All comments have been addressed

2. Is the manuscript technically sound, and do the data support the conclusions?

Reviewer #1: Yes

Reviewer #2: Yes

Reviewer #3: Yes

3. Has the statistical analysis been performed appropriately and rigorously? 

Reviewer #1: Yes

Reviewer #2: Yes

Reviewer #3: N/A

4. Have the authors made all data underlying the findings in their manuscript fully available?

Reviewer #1: Yes

Reviewer #2: Yes

Reviewer #3: Yes

5. Is the manuscript presented in an intelligible fashion and written in standard English?

Reviewer #1: Yes

Reviewer #2: Yes

Reviewer #3: Yes

6. Review Comments to the Author

Reviewer #1: The authors have responded in detail to all reviewers' comments on previous versions of their papers. However, for the question: "Please explain the similarities and differences between the "ZnSS" in the manuscript and the "Z-set" proposed in the research "Z-set Based Approach to Control System Design," the authors did not conduct a sufficient analysis. In addition，the comparative experimental part is still insufficient, and the authors should compare it with the relevant literature from the last two to three years. Finally, the language quality and format of the revised paper still need to be improved.

Reviewer #2: The comments have been addressed.

The paper is original and well-organized with clear examples.

I recommend Accept.

Reviewer #3: All issues have been successfully addressed by authors. The authors have considerably tried to apply my comments, and as a result, the manuscript has significantly improved. Therefore, the manuscript can be accepted in the current form.

7. PLOS authors have the option to publish the peer review history of their article (what does this mean?). If published, this will include your full peer review and any attached files.

Reviewer #1: No

Reviewer #2: No

Reviewer #3: No

---

## [Author Response · Author response to Decision Letter 1]

23 Jun 2022

Dear Editors and Reviewers:

We are very happy for our paper entitled “An approach in medical diagnosis based on Z-numbers soft set” has been approved to revise for closer to the requirements of the journal. We are very grateful to you and the reviewers for their valuable comments and suggestions. We have studied comments carefully and have made correction which we hope meet with approval. Revised portion are marked in yellow in the paper. Besides revised our paper according to the comments and suggestions, we also replenish our paper carefully.

According to the comments given by the reviewers, we divide the comments into several parts and answered for it carefully one by one. The main corrections in the paper and the responds to the reviewer’s comments are in the file "Description of the Revisions for PONE-D-21-34053R1".

---

## [Decision Letter · Decision Letter 2]

15 Jul 2022

An approach in medical diagnosis based on Z-numbers soft set

PONE-D-21-34053R2

Dear Dr. Xiao,

We’re pleased to inform you that your manuscript has been judged scientifically suitable for publication and will be formally accepted for publication once it meets all outstanding technical requirements.

Kind regards,

Ziqiang Zeng, Ph.D.

Academic Editor

PLOS ONE

Additional Editor Comments (optional):

Reviewers' comments:

Reviewer's Responses to Questions

**Comments to the Author**

1. If the authors have adequately addressed your comments raised in a previous round of review and you feel that this manuscript is now acceptable for publication, you may indicate that here to bypass the “Comments to the Author” section, enter your conflict of interest statement in the “Confidential to Editor” section, and submit your "Accept" recommendation.

Reviewer #1: All comments have been addressed

Reviewer #2: All comments have been addressed

Reviewer #3: All comments have been addressed

2. Is the manuscript technically sound, and do the data support the conclusions?

Reviewer #1: (No Response)

Reviewer #2: Yes

Reviewer #3: Yes

3. Has the statistical analysis been performed appropriately and rigorously? 

Reviewer #1: (No Response)

Reviewer #2: Yes

Reviewer #3: I Don't Know

4. Have the authors made all data underlying the findings in their manuscript fully available?

Reviewer #1: (No Response)

Reviewer #2: Yes

Reviewer #3: Yes

5. Is the manuscript presented in an intelligible fashion and written in standard English?

Reviewer #1: (No Response)

Reviewer #2: Yes

Reviewer #3: Yes

6. Review Comments to the Author

Reviewer #1: (No Response)

Reviewer #2: The authors have addressed the required corrections and

updates and we reccomend for the paper to be published.

Reviewer #3: The authors have addressed the point of my concern. I am happy with their corrections. Hence, I would like to recommend this manuscript to be published.

7. PLOS authors have the option to publish the peer review history of their article (what does this mean?). If published, this will include your full peer review and any attached files.

Reviewer #1: No

Reviewer #2: No

Reviewer #3: **Yes: **S. A. Edalatpanah

---

## [Editor Report · Acceptance letter]

11 Aug 2022

PONE-D-21-34053R2 

An approach in medical diagnosis based on Z-numbers soft set 

Dear Dr. Xiao:

I'm pleased to inform you that your manuscript has been deemed suitable for publication in PLOS ONE. Congratulations! Your manuscript is now with our production department. 

Kind regards, 

on behalf of

Dr. Ziqiang Zeng 

Academic Editor

PLOS ONE